# Uncovering the Mechanisms of Adenosine Receptor-Mediated Pain Control: Focus on the A_3_ Receptor Subtype

**DOI:** 10.3390/ijms22157952

**Published:** 2021-07-26

**Authors:** Elisabetta Coppi, Federica Cherchi, Elena Lucarini, Carla Ghelardini, Felicita Pedata, Kenneth A. Jacobson, Lorenzo Di Cesare Mannelli, Anna Maria Pugliese, Daniela Salvemini

**Affiliations:** 1Department NEUROFARBA, Division of Pharmacology and Toxicology, University of Florence, 50139 Firenze, Italy; federica.cherchi@unifi.it (F.C.); elena.lucarini@unifi.it (E.L.); carla.ghelardini@unifi.it (C.G.); felicita.pedata@unifi.it (F.P.); lorenzo.mannelli@unifi.it (L.D.C.M.); annamaria.pugliese@unifi.it (A.M.P.); 2Molecular Recognition Section, Laboratory of Bioorganic Chemistry, National Institute of Diabetes and Digestive and Kidney Diseases, National Institutes of Health, Bethesda, MD 20892-0810, USA; kennethj@niddk.nih.gov; 3Department of Pharmacology and Physiology, Saint Louis University School of Medicine, St. Louis, MO 63104, USA; daniela.salvemini@health.slu.edu

**Keywords:** A_3_ adenosine receptor, neuropathic pain, visceral pain, dorsal root ganglion neurons, Ca^2+^ currents, T cells, interleukin-10, adenosine

## Abstract

Agonists of the G_i_ protein-coupled A_3_ adenosine receptor (A_3_AR) have shown important pain-relieving properties in preclinical settings of several pain models. Active as a monotherapy against chronic pain, A_3_AR agonists can also be used in combination with classic opioid analgesics. Their safe pharmacological profile, as shown by clinical trials for other pathologies, i.e., rheumatoid arthritis, psoriasis and fatty liver diseases, confers a realistic translational potential, thus encouraging research studies on the molecular mechanisms underpinning their antinociceptive actions. A number of pathways, involving central and peripheral mechanisms, have been proposed. Recent evidence showed that the prototypical A_3_AR agonist Cl-IB-MECA and the new, highly selective, A_3_AR agonist MRS5980 inhibit neuronal (N-type) voltage-dependent Ca^2+^ currents in dorsal root ganglia, a known pain-related mechanism. Other proposed pathways involve reduced cytokine production, immune cell-mediated responses, as well as reduced microglia and astrocyte activation in the spinal cord. The aim of this review is to summarize up-to-date information on A_3_AR in the context of pain, including cellular and molecular mechanisms underlying this effect. Based on their safety profile shown in clinical trials for other pathologies, A_3_AR agonists are proposed as novel, promising non-narcotic agents for pain control.

## 1. Introduction

Chronic pain is a highly debilitating condition, disturbing all aspects of our daily experience in social and career-related contexts. The pharmacological tools currently available are sometimes inadequate, or, as in the case of opioids, limited by serious adverse effects [1]. Thus, efforts are being made to pursue research into innovative, non-opioid, pain-relieving compounds.

Many experimental reports have identified adenosine receptors (ARs) as potential targets for acute or chronic pain management. Adenosine is a ubiquitous endogenous neuromodulator whose actions are mediated by four G protein-coupled receptors (GPCR), namely A_1_, A_2A_, A_2B_ and A_3_ receptors (A_1_Rs, A_2A_Rs, A_2B_Rs and A_3_Rs). A_1_Rs and A_3_Rs are coupled to G_i_ members of the G protein family, while A_2A_Rs and A_2B_Rs are G_s_-coupled receptors [2]. The consequent modulation of cyclic adenosine monophosphate (cAMP) levels activates or inhibits a number of signaling pathways, depending on the specific type of cell involved (Figure 1). In some cases, A_2B_R might also couple to G_q_ proteins, as well as A_1_Rs to the G_o_ protein. All ARs are coupled to mitogen-activated protein kinase (MAPK) pathways, including extracellular signal-regulated kinase 1/2 (ERK1/2), p38 MAPK and Jc-Jun-NH2 terminal Kinase (JNK) [3].

A variety of functions are regulated by purines: cardiovascular, respiratory, inflammatory and immune events [4,5,6], as well as neuronal maturation [7,8] and hypoxic damage [9,10]. Of note, adenosine receptors are expressed within the pain-related areas in the peripheral or central nervous system (CNS). It is known that ARs are widely distributed on neurons and glia; hence, considerable interest is focused on the development of selective AR ligands able to control neurological alterations in nervous system diseases [11,12,13,14]. Importantly, local levels of adenosine are significantly enhanced during events of tissue inflammation, stress or trauma, as well as during hypoxia-ischemia [15,16]. Adenosine is a potent anti-inflammatory autacoid that inhibits a number of inflammatory mechanisms, including phagocytosis, the generation of toxic oxygen metabolites, cytokine release and cellular adhesion [17]. Most of these anti-inflammatory effects were first ascribed to the activation of “A_2_Rs” [18], before further investigation attributed the anti-inflammatory effect of adenosine to A_2A_Rs [19], which are widely expressed on peripheral blood and immune cells [20].

Moving in more detail to pain transmission, the first proof of adenosine’s involvement in anti-nociception dates from the 1970s, when the systemic and spinal (intrathecal: i.th.) administration of selective agonists proved effective in pain control. These studies emphasized the role of adenosine A_1_Rs in producing anti-nociception, with some effects ascribed to the A_2A_R subtype [21,22]. Adenosine involvement in peripheral nociception was further confirmed, e.g., the exogenous administration of A_1_R agonists locally to the hind paw of a rat produces anti-nociception in a pressure hyperalgesia model [23], whereas the local administration of A_2_R agonists enhances pain responses [24], an action due to adenosine A_2A_R activation, as confirmed by using the selective agonist CGS21680 [25].

The anti-nociceptive action of A_1_R agonists has been attributed to adenylate cyclase (AC) inhibition and to the decreased production of cAMP in sensory nerve terminals [26,27]. While not visualized directly on sensory terminals, A_1_Rs are present on the cell body of dorsal root ganglion (DRG) neurons [28], and on primary afferent neuronal terminals [29]. In the last decade, investigations into the adenosine-mediated effects in a variety of models for acute or chronic pain were undertaken [30], and the robust protective role of A_1_R emerged [31].

Conversely, the effects of A_2A_R activation on the promotion of cutaneous inflammatory pain have been proposed to result from the stimulation of AC, leading to increased cAMP levels in the sensory nerve terminal [26,27], thus producing opposite effects to those elicited by the anti-hyperalgesic A_1_R subtype. However, the relation between A_2A_Rs and pain has been controversial, with evidence implying either pro-nociceptive or anti-nociceptive activity depending on the receptors’ localization and animal models of pain [30]. Indeed, a relevant A_2A_R-mediated anti-nociceptive effect has been described in a recent study demonstrating that central neuropathic pain evoked by dorsal root avulsion could be reversed by a single intrathecal injection of A_2A_R agonists [32]. The beneficial effects of A_2A_R agonists were associated with reduced reactive gliosis in the CNS. However, A_2A_R antagonists reduced chemotherapy-induced neuropathic pain when administered orally [33]. The discrepancies between the reported effects of A_2A_Rs in pain control could be due to the possible opposing roles that this AR subtype exerts in the periphery (anti-inflammatory effect) versus the CNS (pro-excitatory effect) (for reviews, see [21,22,23,24,25,26,27,28,29,30,31,32,33,34]).

By using an animal model of acute peripheral pain, e.g., subcutaneous glutamate injection into the hind paw of mice, Macedo and co-workers recently confirmed the anti-nociceptive effect of peripheral A_1_R activation by demonstrating that intraplantar (i.pl.) N^6^-cyclohexyl-adenosine (CHA, A_1_R agonist) administration reduced glutamate-evoked nociception. On the other hand, the A_2A_R agonist CGS21680 increased—whereas the A_2A_R antagonist ZM241385 decreased—pain behavior in the same experimental model [35]. Of note, neither the A_2_R agonist, N^6^-[2-(3,5-dimethoxyphenyl)-2-(2-methylphenyl)-ethyl]adenosine (DPMA), nor the A_3_R agonist, 2-hexyn-1-yl-N^6^-methyladenosine (HEMADO), had any effect on glutamate-induced pain. The authors conclude that peripheral A_1_Rs alleviate, while peripheral A_2A_R exacerbate, acute nociception, whereas A_2B_Rs and A_3_Rs do not participate in this particular pain mechanism [35]. Of note, this in vivo experimental model, even if useful to investigate acute pain mechanisms, does not cover the issue of pain chronicization.

Regarding the less studied and less characterized AR subtype in chronic models of pain, A_2B_R, contrasting evidence exists in the literature. On the one hand, A_2B_R activation promotes pain states by increasing the release of interleukin-6 (IL-6) [36,37,38], a pro-inflammatory cytokine also known to cause nociceptor hyperexcitability [36]. On the other hand, the selective A_2B_R agonism, similarly to A_2A_R activation, alleviates the mechanical allodynia induced by different rat models of long-lasting neuropathic pain states, i.e., by long-established chronic constriction injury (CCI; monitored up to 6 weeks after surgery), spinal nerve ligation, or sciatic inflammatory neuropathy [39]. In these experimental paradigms, both A_2A_R and A_2B_R receptor subtypes shared a common intracellular pathway to produce pain relief, consisting in the attenuation of tumor necrosis factor-α (TNFα) production in microglia and astrocytes by protein kinase A (PKA) and/or protein kinase C (PKC) activation [39]. Of note, in the same paper, the authors also confirmed the anti-allodynic effect of selective A_1_R agonism. Interestingly, they found that A_2A_R- or A_2B_R-mediated pain control lasted for a significantly longer time (i.e., up to 6 weeks) than the A_1_R-mediated anti-allodynic effect. This concept could be crucial to gaining insight into the different pain-relieving mechanisms mediated by distinct AR subtypes.

Unfortunately, the clinical translation of A_1_R or A_2A_R agonists for pain relief was hindered by important side effects caused by A_1_Rs being expressed in conducting tissues or A_2A_Rs in vascular smooth muscle, e.g., bradycardia and vasodilation, respectively [40]. Furthermore, data on A_2A_R’s role in pain are elusive and often contradictory [41], thus failing to concretize into a clinical approach.

In contrast, the activation of the A_3_R in humans by potent, selective, and orally bioavailable A_3_R agonists, e.g., IB-MECA (1-deoxy-1-[6-[[(3-iodophenyl)methyl]amino]-9H-purine-9-yl]-N-methyl-β-D-ribofuranuronamide) and its chlorinated counterpart Cl-IB-MECA (2-chloro-N^6^-(3-iodobenzyl)-adenosine-5′-N-methyluronamide), is not associated with cardiac or hemodynamic effects [42], thus pointing to these compounds as potential therapeutics for the treatment of a number of central or peripheral diseases. It is important to note that these A_3_R-selective nucleosides are already in phase 2 and/or 3 clinical trials for autoimmune inflammatory diseases, liver cancer, and non-alcoholic steatohepatitis (see NCT00556894 at https://www.clinicaltrials.gov/ct2/show/NCT00556894?id=NCT00556894&draw=2&rank=1&load=cart; or NCT02927314 at https://www.clinicaltrials.gov/ct2/show/NCT02927314?term=NCT02927314&draw=2&rank=1) [43]. Hence, further attention was focused on this AR subtype for the investigation of innovative pain-relieving strategies, as detailed below.

## 2. A_3_R and Pain

The A_3_R is G_i_-coupled, similarly to the A_1_R, and its activation decreases intracellular cAMP levels, an effect classically related to pain control [44]. However, in the late 1990s, when the first A_3_R-mediated effects on pain were described, it was found that the local application of the A_3_R agonist produced an intrinsic, formalin-like, nociceptive response, and potentiated the algesic effect of low formalin concentrations [44,45]. The pro-nociceptive role of peripheral A_3_R activation was later ascribed to the stimulation of this receptor subtype on mast cells [46], where it is highly expressed and represents one of the most efficacious stimuli to achieve degranulation and histamine release [47]. However, A_3_R activation does not induce degranulation and histamine release in human mast cells [48].

More recently, extensive evidence has highlighted the opposite, i.e., anti-hyperalgesic, effect of A_3_R activation in different rodent models of neuropathic pain. Works performed by the team of Salvemini demonstrated that a single, intraperitoneal (i.p.) injection (at day 7 after injury) of prototypical (IB-MECA, Cl-IB-MECA) or newly synthesized (MRS1898) A_3_R agonists proved effective in relieving neuropathic pain caused by CCI or chemotherapy treatment in rodents [49]. These effects were sensitive to the prototypical A_3_R antagonist MRS1523, but not to the A_1_R and A_2A_R blockers DPCPX and SCH-442416, respectively, nor to the opioid antagonist naloxone [49]. Of note, A_3_R-mediated pain control was achieved after a single dose of IB-MECA administered at the peak of CCI-induced allodynia (on day 7 after nerve ligation: D7), and was not different from that produced by consecutive daily injections of the compound (from D8 to D15). Thus, it was shown that efficient A_3_R-mediated pain relief is achieved after a single acute administration of the agonist, and that the A_3_R does not become tolerant to agonist activation [49]. It is worth noting that A_3_R agonist treatment did not affect the physiological pain threshold in the contralateral paw, indicating a specific anti-hyperalgesic effect in conditions of altered sensitivity that causes the selective alleviation of persistent neuropathic pain states, but without analgesic effects [49,50]. A_3_R agonist-mediated pain relief was absent in A_3_R knock-out (KO) mice [50].

The potent pain-relieving effect of A_3_Rs was further confirmed in mice and rats using a variety of pain models [50,51,52,53,54] and different A_3_R ligands, including innovative, highly selective A_3_R agonists synthesized by Jacobson and coworkers (MRS5841: [55]; MRS7220: [56]; MRS7154: [57]; for a review see [58]), among which are the “first in class” compounds MRS5980 [59] and MRS5698 [60,61] and the water-soluble prodrug MRS7476 [62]. Interestingly, the efficacy of A_3_R stimulation was recently demonstrated against colitis-induced visceral pain [63].

Of note, extracellular adenosine overload during injury or stress conditions can be further increased by inhibiting the enzyme responsible for its extracellular degradation, adenosine kinase (AK) [64]. So, in addition to treatment with AR agonists, the local administration of an AK inhibitor can also modify behaviors produced by an inflammatory stimulus. Accordingly, systemic administration of the AK inhibitor ABT-702, at the peak of CCI-induced pain (D7), significantly increased intraspinal adenosine levels and reversed mechanical allodynia in CCI mice. This effect was partially attenuated by pre- or post-treatment with the selective A_3_R antagonist MRS1523 [50]. Again, no effects of the tested compounds were observed on the mechanical pain threshold in non-injured animals [50]. Similar results were obtained, in the same work, via paclitaxel or oxaliplatin treatment in animals developing peripheral neuropathy, i.e., chemotherapy-induced neuropathic pain (CINP) [50]. This mechanism of action was further confirmed by later work demonstrating that the upregulation of spinal AK exacerbates oxaliplatin-induced allodynia [61] by astrocyte-dependent signaling in the spinal cord, an effect prevented by i.th. injections of the A_3_R antagonist MRS1523 [61].

Taken together, these data indicate that A_3_R-mediated pain control is efficiently achieved in mice and rats [51], and is evident after either systemic (i.p.: [49,50,51,52,53]), oral [49,50,51,52,53,54,55,56,57,58,59,60], subcutaneous [50] or spinal (intrathecal, i.t.: [49,50,51]) administration of A_3_R agonists. All effects were prevented by the spinal administration of the selective A_3_R antagonist MRS1523 [50,51], thus pointing to the predominantly central mode of action of this receptor subtype. However, the exact mechanism(s) of A_3_R-mediated anti-hyperalgesia are still not fully understood and, as stated below, peripheral routes of action have also been described.

## 3. Mechanisms of A_3_R-Mediated Pain Control

Various molecular mechanisms have been proposed to mediate A_3_R-induced pain relief. Janes et al. (2014) demonstrated that A_3_R agonists block the development of chemotherapy-induced neuropathic pain (CINP) by exerting beneficial effects associated with the modulation of spinal neuro-glial communication and neuroinflammatory processes. IB-MECA administered i.p. 15 min prior to each paclitaxel dose attenuated astrocytic activation in the spinal cord by inhibiting NADPH oxidase, and by enhancing redox-sensitive nuclear factor κ-light-chain-enhancer of activated B cells (NFκB) and MAPK kinases such as ERK1/2 and *p*38. This results in a reduced level of the neuroexcitatory/proinflammatory cytokines TNF-α and interleukin-1β (IL-1β), whereas an increase in the neuroprotective/anti-inflammatory interleukin-10 (IL-10) was observed, as well as restored synaptic glutamate homeostasis at the dorsal horns [52] (Figure 2A). These data suggest that the inhibition of an astrocyte-associated neuroinflammatory response in the spinal cord contributes to the protective actions of A_3_R, as later data demonstrate that microglial activation was not observed during the development of oxaliplatin-induced mechano-hypersensitivity in rats [53]. Of note, the authors add important information to this scenario by demonstrating that the A_3_R-mediated inhibition of redox state alterations in the spinal cord during chemotherapy treatment results in a second A_3_R-mediated protective mechanism, e.g., the maintenance of glutamate homeostasis at the synapse, because IB-MECA prevented post-transcriptional alterations in glutamate transporter-1 and glutamine synthetase [52].

Another mechanism was identified by Ford and co-workers, who highlighted the fact that the A_3_R-mediated anti-hyperalgesic effect was completely lost following the intraspinal injection of the γ-aminobutyric acid A (GABA_A_) receptor antagonist bicuculline [51], indicating recruitment of the GABAergic system in the effect. It is known that tonic GABA inhibition in the dorsal horn is inhibited in neuropathic pain states, thus leading to a net neuronal hyperexcitability [65]. Indeed, the authors [51] demonstrated that the deregulation of GABA signaling in CCI rats and mice was due to a reduced GABA synthesis by glutamic acid decarboxylase 65-kDa (GAD65), enhanced extracellular GABA reuptake via the GABA transporter 1 (GAT-1), and by the loss of the K^+^-Cl^−^ co-transporter 2- (KCC2)-maintained Cl^−^ gradient [51], collectively leading to reduced extracellular GABA levels. Interestingly, the systemic or spinal administration of A_3_R agonists (IB-MECA and MRS5698) prevented GAT-1 and GAD65 inactivation, and promoted Cl^−^ gradient maintenance by preserving KCC2 functionality, thus counteracting extracellular GABA decrease and preventing mechanical allodynia (Figure 2B). These effects were blocked by spinal pretreatment with the A_3_R antagonist MRS1523, indicating a central mode of action of the A_3_R in this experimental model [51].

A further spinal mechanism was proposed, e.g., mechanical allodynia induced in rats or mice by CCI or CINP, or cancer-induced bone pain [66]. These were reversed by the A_3_R agonist MRS5698 by reducing the dynamic range of neuron excitability, causing the supraspinal inhibition of nociception through the activation of serotonergic and noradrenergic bulb spinal circuits [50]. Indeed, the authors demonstrated that subcutaneous MRS5698 administered to nerve-injured rats during maximal mechano-allodynia significantly reduced the evocation of wide dynamic-range neuron responses to non-noxious and noxious mechanical, thermal, and electrical stimulation, with peak effects at 1 h post-dosing. Critically, A_3_R agonism did not produce tolerance nor alter nociceptive thresholds in non-neuropathy animals, therefore causing the selective alleviation of persistent neuropathic pain states [50].

Beyond above mentioned effects on the CNS, further evidence from Coppi et al. (2019) shows that peripheral mechanisms could be involved in A_3_R-mediated anti-hyperalgesia, because selective agonists of the receptor (Cl-IB-MECA or MRS5980) applied to isolated DRG neurons decreased neuronal firing and inhibited pro-nociceptive Ca^2+^ currents sensitive to the N-type channel blocker PD732121, an analogue of ω-conotoxin [67] (Figure 3A). It is noteworthy that the selective activation of A_1_Rs by CPA led to a much smaller Ca^2+^ current (I_Ca_) inhibition than Cl-IB-MECA, and the effect of the endogenous agonist adenosine was blocked to a higher extent by the A_3_R-selective antagonist VUF5574 than by the A_1_R-selective blocker DPCPX. Adenosine-mediated I_Ca_ inhibition was completely abolished by a combination of both agonists, demonstrating that A_1_R and A_3_R, but not the A_2_R subtypes, concurrently inhibit N-type I_Ca_ (Cav2.2) in rat DRG neurons, but that A_3_R-mediated effects predominate [67]. These results are in agreement with data from the 1980s showing that adenosine inhibits voltage-dependent Ca^2+^ channels (VDCCs) in isolated mouse DRG neurons [28], an effect that was only partially ascribed to A_1_R activation [68]. Hence, by using the DRG in vitro model, it was demonstrated that A_3_R activation inhibits Ca^2+^ entry into the neurons and action potential firing, suggesting an inhibition of synaptic transmission at the dorsal horn. However, direct evidence for decreased glutamate release into the spinal cord is still lacking. These results are also in line with previous evidence demonstrating that the aberrant expression and/or activity of N-type VDCCs is associated with neuropathic pain [69], and ziconotide, a derivative of ω-CTX, was FDA approved in 2000 (Prialt) for the intrathecal treatment of severe and refractory chronic pain [70,71,72]. However, severe adverse effects (e.g., hallucinations or other psychiatric symptoms) are associated with a direct Ca^2+^ channel block, likely due to their wide expression in the CNS.

A similar mechanism of action, e.g., Cav2.2 inhibition, has been proposed by Lucarini et al. for A_3_R-mediated visceral pain relief in a rat model of experimental colitis reproduced by dinitrobenzenesulfonic acid (DNBS) treatment [63]. In this case, as in the CCI model, a single injection of Cl-IB-MECA, or the more selective A_3_R agonist MRS5980, at the peak of visceral hypersensitivity (day 14 after DNBS injection) was found to be effective in relieving visceral allodynia measured by quantifying the number of abdominal muscle contractions upon intestine distention [63]. Interestingly, the N-type Ca^2+^ channel blocker PD723212 mimicked A_3_R agonist-mediated pain control, thus confirming previous evidence implicating this ion channel in DRG electrical activity [67] (Figure 3a).

An additional peripheral pathway of A_3_R-medited anti-hyperalgesia has recently been identified by Durante and co-workers. Surprisingly, it was found that transgenic Rag-KO mice, lacking T and B cells, are insensitive to the anti-allodynic effects of A_3_R agonist MRS5980, whereas the adoptive transfer of CD4^+^ T cells from wild-type (wt) mice infiltrated the inflamed DRG and restored A_3_R agonist-mediated anti-allodynia [73]. Of note, the adoptive transfer of CD4^+^ T cells from A_3_R-KO or IL-10-KO mice did not restore the A_3_R-mediated anti-hyperalgesic effect, demonstrating, for the first time, that A_3_R activation on CD4^+^ T cells elicits IL-10 release, which, in turn, is responsible for the anti-hyperalgesic effect of MRS5980 [73]. Further downstream mechanisms of the A_3_R-mediated anti-allodynic effect were elucidated in the same work by use of an innovative in vitro model consisting of co-cultures of DRG neurons and T cells isolated from the same animal, either naïve or CCI mice. By this paradigm, CD4^+^ or CD8^+^ T cell infiltration into the ganglion was reproduced in vitro via a circumscribed experimental model, wherein any eventual effects of the cytokine(s) released by the T cells on DRG neurons upon A_3_R stimulation could be evaluated in isolation. Notably, the application of the A_3_R agonist MRS5980 to DRG-CD4^+^ T cell co-cultures significantly reduced the action potential (AP) firing evoked by a depolarizing current ramp and increased the current threshold for AP initiation. The effect was not observed in DRG-CD8^+^ T cell co-cultures, or in DRG neurons cultured alone, demonstrating the selective engagement of A_3_Rs expressed on CD4^+^ T cells [73]. Importantly, the effect of MRS5980 on DRG neuronal firing was also abolished by pre-treatment with an anti-IL-10 selective antibody, thus unequivocally pointing to this anti-inflammatory cytokine as the main effector produced by CD4^+^ T cells upon A_3_R activation, inhibiting neuronal excitability (Figure 3b). The results were not different when either DRG or CD4^+^ T cells were isolated from naïve or CCI animals [73]. Interestingly, the involvement of IL-10 in A_3_R-mediated pain control was already evident in a previous work, where attenuating IL-10 signaling in oxaliplatin-treated rats was achieved with an intrathecal neutralizing IL-10 antibody, or in IL-10-/- mice [61].

Interestingly, a recent paper outlined that the pain-relief properties of some tricyclic antidepressants, such as amitriptyline, one of the most widely prescribed antidepressants for neuropathic pain management [74], might be at least partly ascribed to AR activation, since it is blocked by caffeine pre-treatment [75,76]. In particular, it was demonstrated that A_3_R antagonists prevent amitriptyline-induced anti-hyperalgesia in CCI rats by preventing the decrease in pro-inflammatory cytokines produced by spinal nerve ligation and by the phosphorylation of ERK1/2 and cAMP response element-binding protein (CREB) [77]. This suggests that A_3_R activation might facilitate the balance between pro-inflammatory and anti-inflammatory pathways in favor of the latter.

Interestingly, a certain degree of sexual dimorphism in A_3_R-mediated pain control has recently been outlined. Indeed, differences in the effects of A_3_R agonists in male and female rodents with paclitaxel-, oxaliplatin- or bortezomib-induced peripheral neuropathy have been reported. The study shows that the A_3_R agonist MRS5698 failed to prevent bortezomib-induced neuropathic pain in female, but not male, rodents [78]. Of note, morphine and duloxetine, both clinical analgesics, were equally effective in both sexes, demonstrating that, on one hand, different chemotherapeutic treatments engage distinct intracellular pathways to produce mechanical allodynia, and, on the other, the lack of response to bortezomib-induced hyperalgesia in females is specific to A_3_R ligands. In the same work, the authors suggested that the lack of effect of MRS5980 in female rats was likely due to the absence of A_3_R overexpression in the spinal cord after bortezomib treatment, in contrast to male rats, in which A_3_R is overexpressed [78].

Finally, A_3_R agonists could be attractive co-therapeutics with opioids. Indeed, the prevention of morphine-induced AK overexpression in the dorsal horn of the spinal cord provided a >90% attenuation of antinociceptive tolerance in CCI mice, an effect mimicked by the selective A_3_R agonists IB-MECA and MRS5698 [79]. Interestingly, and in line with the above results, the beneficial effects of A_3_R stimulation in mechanical allodynia were associated with the inhibition of the NOD-like receptor pyrin domain-containing 3 inflammasome in the spinal cord, leading to decreased IL-1β production and increased IL-10 levels [79]. Furthermore, A_3_R activation following treatment with levels of A_3_R agonist (i.th.) that had no effect on nociceptive threshold in morphine-naïve rats potentiated/restored antinociception (tail flick assay) in morphine-tolerant rats [80].

## 4. Conclusions

The modulation of A_3_Rs induces potent anti-hypersensitive effects in diverse preclinical models of chronic pain. Nevertheless, the mechanism by which this AR subtype exerts anti-hyperalgesic and anti-allodynic effects is still to be clarified, and before now, both peripheral (CD4^+^ T cell-mediated production of anti-inflammatory IL-10 or DRG neuronal inhibition) and central (spinal astrocyte reactivity inhibition, increased GABA release in the spinal cord, inhibition of spinal and supraspinal activation of serotonergic and noradrenergic circuits) effects have been described, with some gender-specific effects outlined in particular cases (e.g., bortezomib-induced neuropathic pain). Of note, the efficacy of A_3_R ligands for the pharmacological control of chronic pain states is particularly relevant because it is devoid of tolerance effects (in animal models) or significant side effects (in animals and humans). However, the clinical relevance of A_3_R-mediated pain control still needs to be demonstrated, and, even if results from clinical trials for other pathologies encourage the use of these ligands for their safe profile, future work will help us elucidate the feasibility of A_3_R-based treatments in humans.

## Figures and Tables

**Figure 1 ijms-22-07952-f001:**
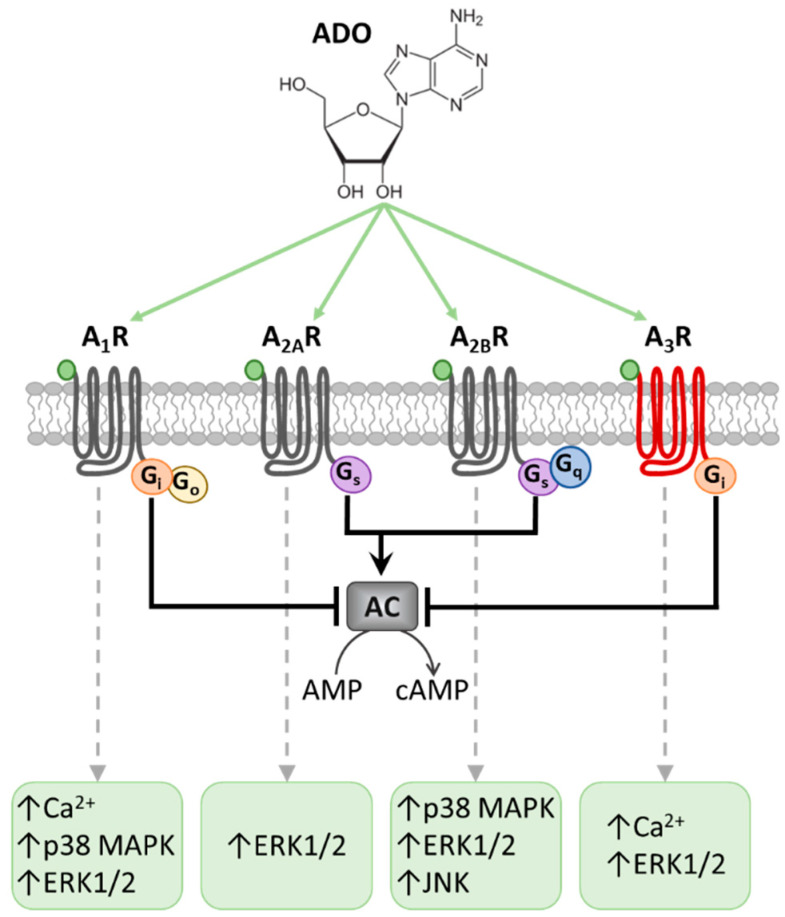
Adenosine receptors and the main transduction pathways involved in their activation. Schematic representation of G protein-coupled A_1_, A_2A_, A_2B_ and A_3_ receptor (A_1_R, A_2A_R, A_2B_R and A_3_R) subtypes activated by extracellular adenosine (ADO), and the main intracellular pathways involved. A_2A_Rs and A_2B_Rs are coupled to the Gs protein, which leads to adenylyl cyclase (AC) activation and cyclic AMP (cAMP) increase. On the other hand, A_1_R and A_3_R are coupled to the G_i_ protein that inhibits AC and reduces cAMP. In some districts, A_2B_Rs are also coupled with G_q_ proteins and A_1_R with Go, which stimulate Ca^2+^ release from intracellular stores. All adenosine receptors are coupled to mitogen-activated protein kinase (MAPK) pathways, including extracellular signal-regulated kinase 1/2 (ERK1/2), p38 MAPK and Jc-Jun-NH2 terminal Kinase (JNK).

**Figure 2 ijms-22-07952-f002:**
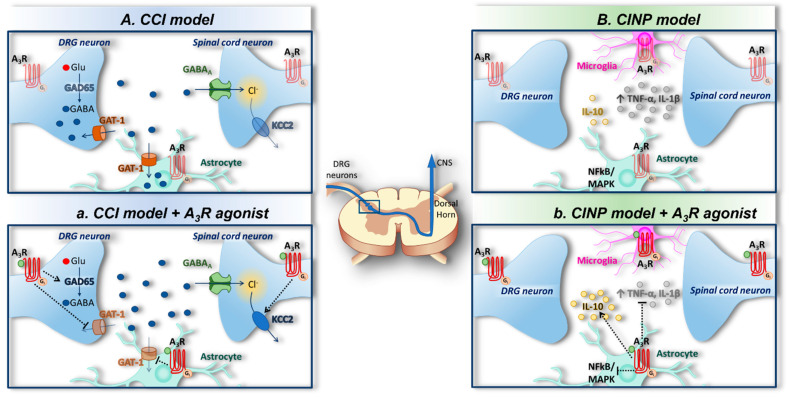
Adenosine A_3_ receptors and spinal mechanisms of pain control. (**A**) Chemotherapy-induced neuropathic pain (CINP) is characterized by high neuroexcitatory/proinflammatory cytokine (TNF-α and IL-1β) release and the activation of NFκB and MAPK in the spinal dorsal horn. (**a**) Glial A_3_Rs inhibit NFκB and MAPK activation, thus leading to a decreased release of pro-inflammatory TNF-α and IL-1β and an increased release of anti-inflammatory interleukin-10 (IL-10) [52]. (**B**) Chronic constriction injury (CCI), an animal model of neuropathic pain, induces deregulation of γ-aminobutyric acid (GABA) signaling via a reduction in GABA synthesis by glutamic acid decarboxylase 65-kDa (GAD65), and enhanced GABA reuptake via the GABA transporter 1- (GAT-1) and the loss of Cl^−^ gradient by K^+^-Cl^−^ co-transporter 2 (KCC2) inhibition, leading to an overall reduction in GABA at the synapse. (**b**) A_3_ receptor (A_3_R) activation prevents CCI-induced GAT-1 enhancement and GAD65 inhibition as well as KCC2 loss of function [51], augmenting extracellular GABA levels.

**Figure 3 ijms-22-07952-f003:**
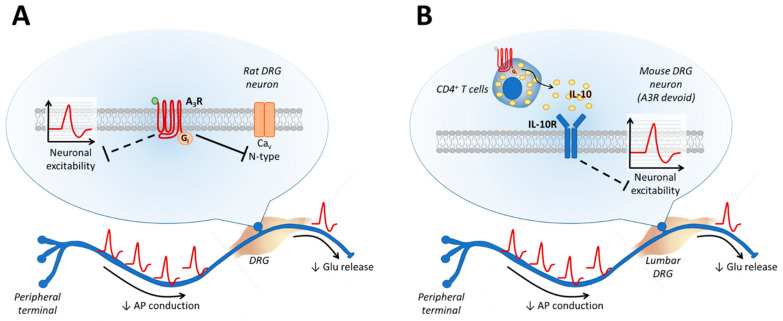
Adenosine A_3_ receptors and peripheral mechanisms for pain control. (**A**) A_3_ receptors (A_3_Rs) expressed on rat DRG neurons decrease action potential (AP) firing by blocking neuronal excitability. The same receptor subtype also inhibits N-type voltage-gated calcium channels (Ca_V_), thus reducing glutamate (Glu) release at the synapse [67]. (**B**) In a mouse model of CCI, A_3_Rs expressed on CD4^+^ T cells, but not on mouse DRG neurons in this rodent species, promote interleukin-10 (IL-10) release, which, by activating IL-10 receptors (IL-10R) on DRG neurons, reduces neuronal excitability [73].

## Data Availability

Not applicable.

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
