# Peer review of "Uncovering the Mechanisms of Adenosine Receptor-Mediated Pain Control: Focus on the A3 Receptor Subtype"

_ijms, 2021, doi:10.3390/ijms22157952_

Round 1

Reviewer 1 Report

The authors comprehensively reviewed the available research on adenosine receptor mediated pain control, concentrating on A3 receptor pathways.

The paper is well written and presents the topic in a well-organized, logical manner. It contributes to the filed by reviewing the available literature.

Major comments:

  1. As adenosine receptor research is plagued by agonist specificity issues, this problem needs to be mentions. None of the AR agonists/antagonists are truly specific for theirs target receptors. Even when the specificity is decent, higher (yet still physiologically achievable) concentrations will trigger the response from more than one AR subtype, and often from all of them. Please include a comment highlighting this issue (probably somewhere around line 110).

  1. I would appreciate a more critical approach to presented research, reviewing also the applied methodology and assessing the validity of the source material.

  1. The title is not inappropriate, but moderately misleading - the bulk of the article reviews adjacent subjects, such as A2 receptor pain-related pathways.

  1. Please make all notations of adenosine receptor subtypes cohesive, now it’s sometimes A2AR, sometimes A2A

  1. Please make sure that all of the abbreviated terms are introduced then they first appear (ICa – line 261), only once (intrathecal, line 71 and line 194), and only when necessary. A list of abbreviations is always nice.

  1. Figure 1: higher resolution is necessary

  1. Figure 1: What is the EC50 on the graph regarding? It’s not defined, nor mentioned anywhere. As EC50 is interaction-, assay- and applied conditions- specific, it obviously needs to be either removed from the graph or fully defined.

  1. Line 138: please give the trial identifiers, when mentioning ongoing clinical trials.

  1. Defining the Authors own work as “pioneer” seems less then elegant. Also, please consider unifying the honorifics usage, so not only Prof. Salvemini and Dr. Jacobson are enjoying their academic titles. My recommendation would be to revert to generally accepted form, as used through the rest of the article (for example line 201, and 231), without extra special consideration given to the article’s co-authors. (As a side note – “Dr” – no dot.)

  1. Please make sure all of the terms traditionally denoted in cursive are as such – “in vivo”, “in vitro”.

  1. Figures 2 and 3 are very nice. There are minor visibility difficulties with the labelling in sublet colours (IL-10, TNFalpha, IL-1beta, GAD65, GAT-1). “Macroglia” is nearly invisible.

Minor comments:

  1. Line 16: confer (no “s”)
  2. Line 23: space after the dot, before “The”
  3. Line 25: safety profile
  4. Lines 32 and 33: Sentence starting “Poor…” needs to be rewritten
  5. Line 30: unnecessary dot before “Introduction”
  6. Line 337 and 338: sentence needs to be rewritten
  7. Line 349: unnecessary dot before “Conclusions”
  8. There are obvious formatting problems with the references, some have titles denoted all in caps, for example ref 18, 21, 23, 65. Please correct.

Author Response

Reviewer 1

Comments and Suggestions for Authors

The authors comprehensively reviewed the available research on adenosine receptor mediated pain control, concentrating on A3 receptor pathways.

The paper is well written and presents the topic in a well-organized, logical manner. It contributes to the filed by reviewing the available literature.

Major comments:

  1. As adenosine receptor research is plagued by agonist specificity issues, this problem needs to be mentions. None of the AR agonists/antagonists are truly specific for theirs target receptors. Even when the specificity is decent, higher (yet still physiologically achievable) concentrations will trigger the response from more than one AR subtype, and often from all of them. Please include a comment highlighting this issue (probably somewhere around line 110).

Response 1. We thank the reviewer for rising this point that, indeed, has been problematic, historically, and has challenged medicinal chemists and pharmacologists in the field. However, many of the newer ligands have sufficiently high selectivity for a given adenosine receptor subtype that ambiguity is avoided. A clear example of a major improvement in selectivity is for the recent generation of A3 receptor agonists containing methanocarba ring in place of ribose combined with a 2-arylethynyl substituent, including MRS5698, MRS5980 and other congeners, on which are based a number of data described in the text. These compounds have nM affinity at the target A3 receptor but negligible affinity (typically >10 µM) at the other three adenosine receptors and represent an important achievement in the study of purinergic signalling.

2. I would appreciate a more critical approach to presented research, reviewing also the applied methodology and assessing the validity of the source material.

 Response 2. We thank the reviewer for rising this important point. The following sentences were added to provide a more critical point of view.

Page 4 lines 141-143: “Of note, this in vivo experimental model, even if useful to investigate acute pain mechanisms, does not cover the issue of pain chronicization.”

Page 8 lines 308-313: “Hence, by using the DRG in vitro model, it was demonstrated that A3R activation inhibits Ca2+ entry into the neurons and action potential firing, suggesting an inhibition of synaptic transmission at the dorsal horn. However, direct evidence for decreased glutamate release into the spinal cord is still lacking.”

3. The title is not inappropriate, but moderately misleading - the bulk of the article reviews adjacent subjects, such as A2 receptor pain-related pathways.

Response 3. We changed the title to include a wider range of adenosine receptors, as suggested, from: “Uncovering the mechanisms of A3 adenosine receptor-mediated pain control” to: “Uncovering the mechanisms of adenosine receptor-mediated pain control: focus on the A3 receptor subtype

4. Please make all notations of adenosine receptor subtypes cohesive, now it’s sometimes A2AR, sometimes A2A

Response 4. We apologise for some incongruence in adenosine receptor notations. We now turned all A2AR to A2AR, A3R to A3R, etc. 

5. Please make sure that all of the abbreviated terms are introduced then they first appear (ICa – line 261), only once (intrathecal, line 71 and line 194), and only when necessary. A list of abbreviations is always nice.

Response 5. We corrected abbreviations to be introduced only once, when necessary, and at first appearance. Furthermore, a list of abbreviation has been added. 

6. Figure 1: higher resolution is necessary

Response 6. We provided a higher resolution version of Figure 1. 

7. Figure 1: What is the EC50 on the graph regarding? It’s not defined, nor mentioned anywhere. As EC50 is interaction-, assay- and applied conditions- specific, it obviously needs to be either removed from the graph or fully defined.

Response 7. As suggested, we removed EC50 values from Figure 1.

8. Line 138: please give the trial identifiers, when mentioning ongoing clinical trials.

 Response 8. Trial identifiers have been added in the text, as suggested. Page 5 Lines 171-173: “(see www.clinicaltrials.gov; NCT00556894; NCT02927314)”.

9. Defining the Authors own work as “pioneer” seems less then elegant. Also, please consider unifying the honorifics usage, so not only Prof. Salvemini and Dr. Jacobson are enjoying their academic titles. My recommendation would be to revert to generally accepted form, as used through the rest of the article (for example line 201, and 231), without extra special consideration given to the article’s co-authors. (As a side note – “Dr” – no dot.)

 Response 9. The word “pioneer” has been deleted (page 5 line 189). We also homologated all names cited in the text by omitting academic titles.

10. Please make sure all of the terms traditionally denoted in cursive are as such – “in vivo”, “in vitro”.

 Response 10. We now checked that terms traditionally denoted in cursive are as such (i.e. “in vitro” at page 9 lines 338 and 341).

11. Figures 2 and 3 are very nice. There are minor visibility difficulties with the labelling in sublet colours (IL-10, TNFalpha, IL-1beta, GAD65, GAT-1). “Macroglia” is nearly invisible.

Response 11. We agree that Figures 2 and 3 could be improved and we modified colours accordingly.

Minor comments:

  1. Line 16: confer (no “s”). Amended
  2. Line 23: space after the dot, before “The”. Amended
  3. Line 25: safety profile. Amended
  4. Lines 32 and 33: Sentence starting “Poor…” needs to be rewritten. The sentence has been rewritten as follows: “Pharmacological tools available to date are sometimes inadequate or, as in the case of opioids, limited by serious adverse effects.” (Page 2 lines 63-64).
  5. Line 30: unnecessary dot before “Introduction”. Amended
  6. Line 337 and 338: sentence needs to be rewritten. The sentence has been modified as follows: “Indeed, differences in the effect of A3R agonists in male and female rodents in paclitaxel-, oxaliplatin- or bortezomib-induced peripheral neuropathy have been reported.” (Page 9 lines 370-373).

For the sake of clarity, the following sentence was also reformulated: “In the same work, the authors suggested that the lack of effect of MRS5980 in female rats was likely due to the absence of A3R overexpression in the spinal cord after bortezomib treatment, in contrast to male rats in which A3R is overexpressed”. (Page 9 lines 378-381)

7. Line 349: unnecessary dot before “Conclusions”. Amended

8. There are obvious formatting problems with the references, some have titles denoted all in caps, for example ref 18, 21, 23, 65. Please correct. Amended.

Reviewer 2 Report

This review focus on adenosine receptors role in pain control, an interesting topic , that has been already covered in many review articles. In particular, in this paper the authors provide a comprehensive review on the role of A3 adenosine receptor in controlling pain, describing the preclinical evidence in support of its therapeutic potential. The manuscript is well written. Some typos errors should be corrected.

Here are some minor comments, to be addressed, before it can be accepted for publication in the journal.

  1. Resolution of Figure 1 is poor.
  2. Lane 101 Macedo and co-workers instead of Nascimento and co-workers (Ref #35)

Author Response

Reviewer 2

Comments and Suggestions for Authors

This review focus on adenosine receptors role in pain control, an interesting topic , that has been already covered in many review articles. In particular, in this paper the authors provide a comprehensive review on the role of A3 adenosine receptor in controlling pain, describing the preclinical evidence in support of its therapeutic potential. The manuscript is well written. Some typos errors should be corrected.

We thank the Reviewer for his/her positive comments on our work.

Here are some minor comments, to be addressed, before it can be accepted for publication in the journal.

  1. Resolution of Figure 1 is poor.

Response 1. We provided a higher resolution version of Figure 1. 

2. Lane 101 Macedo and co-workers instead of Nascimento and co-workers (Ref #35)

Response 2. Amended (line 132 of revised manuscript)
